# Implementation and Evaluation of Open-Source Hardware to Monitor Water Quality in Precision Aquaculture

**DOI:** 10.3390/s20216112

**Published:** 2020-10-27

**Authors:** Rafael Apolinar Bórquez López, Luis Rafael Martinez Cordova, Juan Carlos Gil Nuñez, Jose Reyes Gonzalez Galaviz, Jose Cuauhtemoc Ibarra Gamez, Ramon Casillas Hernandez

**Affiliations:** 1Departamento de Ciencias Agropecuarias y Veterinarias del Instituto Tecnológico de Sonora, 5 de Febrero 818 Sur, Cd. Obregón 85000, SON, Mexico; rafael.borquezl@itson.edu.mx (R.A.B.L.); jcgil0612@gmail.com (J.C.G.N.); jose.ibarra@itson.edu.mx (J.C.I.G.); 2Departamento de Investigaciones Científicas y Tecnológicas de la Universidad de Sonora, Boulevard Luis Donaldo Colosio Colosio s/n, Hermosillo 83000, SON, Mexico; luis.martinez@unison.mx; 3CONACYT—Instituto Tecnológico de Sonora, 5 de Febrero 818 Sur, Cd. Obregón 85000, SON, Mexico; jose.gonzalez@itson.edu.mx

**Keywords:** precision aquaculture, arduino, water quality, low-cost

## Abstract

Precision aquaculture is a new field that allows farmers to have better control over aquaculture processes, facilitating decision-making and improving efficiency. The implementation and evaluation of a low-cost water-quality monitoring system based on open-source hardware, which is easy to rebuild for scientific applications, is reported in this paper. The proposed system measures temperature, dissolved oxygen, and pH, taking records and sending information through a wireless protocol (ZigBee) to a graphical user interface which can display information numerically and graphically, as well as simultaneously storing the information in a database. These variables are very important for aquaculture, as they have a direct influence on critical culture parameters such as growth and survival. Although it is a low-cost system, it offers good quality data and demonstrates efficiency for its use in precision aquaculture.

## 1. Introduction

Over the last four decades, aquaculture has become the fastest-growing agroindustry worldwide [1]. The global production of fish continues to grow slower than the population (mainly as fishing remains almost stable, at around 80 million MT). In this context, aquaculture has contributed more day by day, with a production of around 80 million MT (110 million MT, if aquatic plants are considered) having been reported by the FAO in 2018, with an associated value of approximately 243.5 billion dollars. This has increased the availability of aquatic food for human consumption from 10 kg per capita in 1960 to more than 20.2 kg per capita in 2015 [2]. The accelerated growth of aquaculture has made the necessary design and development of production systems with new technologies into a field known as “precision aquaculture” (or PAq). These technologies are capable of assisting aquafarmers in decision-making, including in intensified systems [3]. PAq can be considered as the application of information technologies (software and hardware) into the field of aquaculture biology, in order to provide support to the production systems. In developing countries, it is very relevant to facilitate access to such technologies, adapting them to the regional situation at accessible costs.

One of the most important aspects to consider in intensive aquaculture is the maintenance of water quality, as it is directly related to the adequate development of the farmed organisms. Poor water quality may be responsible for creating stress conditions, which can affect feed consumption, growth, survival, and reproduction. Thresholds or adequate levels of water-quality parameters depend on the type of culture [4]. The monitoring of physical and chemical variables, such as dissolved oxygen, temperature, and pH, in the water column is vital to maintain adequate conditions and to avoid undesirable situations which may lead to the collapse of aquaculture systems [5]. It has been well-documented that some aquatic species are highly sensitive to drops in dissolved oxygen levels, abrupt variations in temperature, and changes of pH. For those reasons, most instrumentation systems include sensors for these parameters, not only for monitoring them, but also to have early information about potential contaminants in the water [6]. Dissolved oxygen (DO) is a very important and limiting factor in aquaculture systems, as all aquatic organisms (except for some bacteria) need a source of DO to live and develop. For this reason, the concentration of DO in the water column is one of the most important aspects to manage for suitable aquaculture. Low levels of DO can affect the feed consumption and, in extreme cases, can cause mass mortalities of aquatic organisms [7,8,9]. The pH in freshwater ranges around 7, while that in marine waters is around 8. For shrimp farming, the recommended pH for optimal growth is from 7.0 to 9.0 [7]. When the pH is under 7 (acid), adverse effects on gill function and growth may result. If the level decreases under 4, acid death occurs; meanwhile, at levels over 11.0, alkaline death is possible. Temperature is another very important parameter for aquatic organisms, mainly for those which are poikilotherms, as their corporal temperature varies directly as function of environmental temperature. Wide variations in internal temperature can have significant effects on physiology and metabolism and, consequently, on feed consumption, growth, reproduction, and so on [9].

For the development of instrumentation systems, platforms known as open source hardware (OSH) have become widely available, which are new tools for the implementation of electronic projects. The Arduino, for instance, is a commercial platform which is very popular in the student community, as it permits the development of a suite of automation projects and, additionally, can save time. It was created in 2005 by the Interaction Design Institute Ivrea, Italy. OSH have become increasingly more utilized, due their versatility, in applications such as robotics, automation, precision agriculture, and many others [10,11,12]. On the other hand, wireless sensors networks (WSN) have become very useful tools to process information in an efficient manner for laboratories, saving time in analyzing samples [13]. At present, WSNs are used as a reliable tool for the monitoring of water-quality parameters (e.g., dissolved oxygen, temperature, and pH) in real time [14,15,16]. With the design of monitoring systems for water quality in aquaculture (including sensors for the aforementioned parameters), based on WSN with the transmission protocol ZigBee, it is possible to analyze and manage information of the environment around the organisms, which can help the farm manager to make decisions regarding aspects such as the increase of culture density (based on water-quality improvement); supplying the feed ration more adequately, or improving the feed conversion and the growth ratios (based on the interactions between consumption, temperature, and oxygen) [17]; maintaining the water-quality parameters more effectively [18]; reducing the eutrophication and hyper-nutrification produced by effluents; preventing the stress and diseases of farmed organisms [19]; and reducing the need for employees [20]. Low-cost (LC) monitoring systems for aquaculture are a priority in developing countries, as they can reach most people and provide benefits. Considering the aforementioned reasons, we decided to carry out this study. In this work, the development of a low-cost open-source hardware (LC OSH) for water-quality monitoring using simple electronics components—an RTD (resistance thermometer), pH sensor, and dissolved oxygen electrode—is studied. The present investigation is focused on the implementation and evaluation of a water-quality monitoring system using LC OSH for application in PAq, capable of making graphs and storing data for further processing and management. In such a way, this comprehensive study can allow for a simple design for a portable device fabrication.

## 2. Materials and Methods

The low-cost water-quality monitoring system was designed and evaluated at the Aquaculture Laboratory of the Department of Agricultural and Veterinary Science of the Instituto Tecnológico de Sonora, sited in Ciudad Obregón, Sonora, México (27.48° 29′ 2.1″ N, 109.93° 56′ 1.9″ W).

Briefly, the system consists of an open-source platform for hardware development with a network of wireless sensors of temperature, dissolved oxygen, and pH sensors. The system was adapted using software on a desktop computer using a serial port based on a depurator (USB serial converter). Figure 1 shows the model used for the monitoring of water-quality variables. The information generated by the sensors was read every minute and stored every 5 min; complete measurement of all three variables took about 3 s.

### 2.1. Hardware

The hardware design was based on the platforms Iteaduino/Arduino. The Arduino chip consists of a microcontroller with easy-to-use hardware and software. These platforms have the particularity of being open-code systems which work in a programing language similar to c++ [21]. In the present study, the platforms Iteaduino/Arduino version Mega were used; with an ATmega 2560 chip (specification sheet 2560, Atmel). Using open code from Arduino, the chip utilized was 100% compatible, having the same characteristics and with the same number of i/o pins, a wide pulse modulator with 8-bit resolution, 16 MHz oscillator crystal, universal serial communication port (UART), and 256 KB of flash memory, among other specifications. Figure 2 shows a picture of the platform and the depurator used. The depurator was manufactured by Atlas Scientific and its function was introducing the information to the computer and receiving instructions from the software; it works as a simple converter (from USB to serial). With the depurator, it is possible to power external modules which require 5 or 3 volts of direct current. The pins Rx and Tx shown in Figure 2 are connected to a wireless transceiver ZigBee, which is a key part of the WSN. The transceiver was fed 3 V from the depurator module. For appropriate functioning, the sensors were calibrated as recommended by the manufacturers, comparing them with commercial equipment limited to continuous automatic monitoring (data logging). Electrodes were installed completely vertically with the use of a two-inch PVC tube and fastened with a plastic belt to ensure the immobility of sensors and at the same measurement point, as shown in Figure 3.

### 2.2. Sensors

#### 2.2.1. Dissolved Oxygen (DO)

In the present study, a kit sensor for DO was used. It consists of an analogue passive galvanic dissolved oxygen probe and an embedded circuit with the capacity for two communication protocols: universal asynchronous port receptor/transmitter (UART) and I^2^C inter-integrated circuit. The DO sensor can read in a range from 0.01 to +35.99 mg L^−1^ with a precision of ±0.2 mg L^−1^. The kit can compensate for temperature, salinity, and atmospheric pressure, its operation voltage is from 3.3 V to 5.0 V, and it is compatible with any microprocessor supporting UART communication. It needs to be calibrated once a year, depending on the operation conditions and use time. The calibration was done using a reference solution with 0 mg L^−1^ of DO [22].

#### 2.2.2. PH

The analogue passive pH electrode connected to the embedded circuit for monitoring the pH in the system was similar to that used for DO, in that it supports both the UART and I^2^C communication protocols. It can read in a range from 0.001 to 14.000 with a precision of ± 0.02; additionally, it is possible to read temperature dependently or independently and, in this case, to have a better precision and accuracy, it was programmed to consider the dependence of pH on temperature [23]. Its operation voltage is from 3.3–5.0 V and it is compatible with any microprocessor supporting UART communication. Calibration may be done once a year and it has a flexible calibration protocol with one, two, or three points, using well-known standard solutions. For the present study, it was calibrated at three points—4.0, 7.0, and 10.0—to ensure precise measurement [22].

#### 2.2.3. Temperature

In order to monitor this variable, a digital sensor tolerant of aggressive environments and any type of climatic condition (considering international standards and protection grade) was used (IP 68). These characteristics make the sensor adequate for use while submerged in either fresh or marine water. It reads in a range from –20 °C to 133 °C with a precision of ± 1 °C. Its operation voltage is 3–5 V direct current, as in the case of the other sensors. Communication with the microcontroller was carried out through the protocol UART configured for a transmission velocity of 9600 [22].

### 2.3. Software

The proposed water-quality monitoring system reads the values of the sensors through a graphical user interface (GUI) software, which is responsible for calling them and establishing a water-quality database. This permits access to analyses of future information, helping to recognize the conditions of the culture. The data reads are also graphed. The GUI was designed and programmed in the C# language, with the objective of having user-friendly, easy-to-use software. When the program is run, the connection is first carried out and the WSN is identified. When a wireless signal is available, the data is sent to the computer to be captured by the program and displayed in a numerical way, graphed, and stored, as shown in the flow diagram of Figure 4.

### 2.4. Performance Assessment

The LC reference equipment was installed in an 8 L aquarium with marine water and an air pump (RESUN AC-9602). This small system was chosen, in order to have better control and to ensure the same water quality throughout the aquarium. The reference equipment consisted of a pH meter (HANNA Instruments HI98128) and a multiparameter sensor for DO and temperature (YSI 551). With this equipment, 10 reads were taken per day (at different hours, in order to have randomized samples). To evaluate the efficiency and reliability of the system, four tests were performed: validation, operation continuity, reproducibility, and reliability. The validation test was done by comparison of the results randomly recorded by the commercial equipment with those obtained with the LC by means of an analysis of variance (ANOVA) of repeated measurements (*P* = 0.05, RMANOVA) using the software Sigma Plot for Windows version 12.0. The operation continuity test was done by maintaining the equipment in continuous operation for a long time, in order to evaluate its performance at present and provide a basis for further improvements. The reproducibility test consisted of the evaluation of three similar LC systems, in order to confirm the reproducibility of their reads and to detect whether significant differences were found among them, by means of one-way ANOVA (*P* = 0.05). Finally, the reliability test consisted of an evaluation of the sensitivity, resolution, precision, and accuracy of the equipment, analyzing each variable independently. For the sensibility analysis, Equation (1) (proposed by [24]) was used, where m is the dependent variable, determined by the least squares method, and σ is the standard deviation. The precision of any instrument is determined by the standard deviation of the reads for each variable and the deviation coefficient. Equations (3)–(5) were used to evaluate the precision of our equipment, while Equation (6) determined the accuracy of the data capture. Figure 5 shows the normal distribution of the reads, which indicates the characteristics of the instrument [24].
(1)$$\gamma =\frac{m}{\sigma }$$
(2)$$m=\frac{\mathrm{Sxy}}{Sxx}$$
(3)$$\sigma =\sqrt{\frac{\sum_{i=1}^{N}{\left(xi-\mu \right)}^{2}}{N}}$$
(4)$$\mu =\frac{\sum_{i=1}^{N}xi}{N}$$
(5)$$CV=\frac{\sigma }{\mu }*100$$
(6)$$A=\frac{\mu -x_{t}}{x_{t}}*100$$

## 3. Results 

The data of the validation test showed no significant differences in any of the measured variables. The mean temperature of the reference equipment was 24.02 °C, while that in the proposed system was 24.05 °C, giving a non-significant difference of 0.03 °C. In the case of dissolved oxygen, the trend was similar, with a mean value of 7.14 mg L^−1^ in the reference equipment and 7.13 mgL^−1^ in the proposed system. For pH, the difference was a little greater (0.06), with a mean of 8.18 in the reference equipment and 8.24 in the proposed system. Table 1 shows the values recorded for each variable in the reference equipment and the proposed system. The highest coefficient of variation was observed for DO, with 17.72% to 18.19% of variation, which indicates that those sensors were a little more unstable, due to the water characteristics and the effect of temperature on this parameter. The pH standard deviation in the proposed system was ± 0.11; a similar value of ± 0.21 when using a wireless water-quality monitor has been reported [25]. 

Table 2 shows the results of repetitive tests; no statistically significant differences were found between them (*P* < 0.05).

A list of the utilized parts is presented in Table 3. It was fulfilled with the aim of developing a system with a low budget. The proposed system had an overall cost of 455€, in comparison to 1030€ with a similar system on the market (commercial price from personal communication with a local store); this represents a difference of 56% (580€). With the proposed system, there is a cost of approximate 150€ per variable measured. The proposed water-quality monitoring system, therefore, meets the requirement of a low-cost system, due to its price. Furthermore, it enables data logging. As can be seen from a comparison with commercially available equipment, its resolution and accuracy are sufficient for qualitative water-quality monitoring. In general, there is no other related equipment to compare with that presented in this study; however, it could be compared to other recent low-cost systems made with similar hardware, where costs can be compared [26,27,28].

Information stored in the database can be processed and displayed as a graph, from which the user can understand the status of an aquaculture system in a better way, and can correct problems or make decisions as soon as possible. Figure 6 shows these results in a graphical way, showing that the application of both systems in aquaculture is feasible and gives reliable information. The comparison between the reference equipment against the proposed LC OSH system follows the same pattern: the differences in the median values between groups are not great enough to exclude the possibility that the difference is due to random sampling variability and statistically significant differences (*P* < 0.05) were determined by Student’s t-tests.

For Figure 7a–c, *n* = 38 samples were taken completely at random and a positive linear correlation was obtained with the reference equipment, where it can be seen that the measurement of dissolved oxygen (Figure 7a) showed very good agreement, obtaining r^2^ = 0.81; the LC-OSH system average was 4.84 mg L^−1^, while that for the reference equipment was 4.82 mg L^−1^, representing a difference of only 0.5%. In a similar way, the measurement of pH obtained r^2^ = 0.72, with an average of 8.26 for the low-cost proposed system and 8.13 for the reference equipment, a difference of 1.5% between their averages. In the case of temperature, r^2^ = 0.97 was obtained, with an average temperature of 26.13 °C for the low-cost proposed system and 26.09 °C for the reference equipment; thus, their difference was 0.2%.

Figure 8 shows the results of the operation continuity test (reads of DO, temperature, and pH over 5 days), which demonstrate the sturdiness of the proposed LC OSH system, as the equipment worked continuously over 30 days without any problems. This is a very important finding as, in aquaculture, the production cycles are very long (up to one year), and the equipment must work without interruption. The maximum value recorded for dissolved oxygen was 8.3 mgL^−1^ and the minimum was 8.0 mgL^−1^; therefore, it can be noted that the dissolved oxygen was always in a ± 0.3 (maximum–minimum) range, where the oxygen source (blower) was never shut down. It can be noted that, as the temperature increases, oxygen decreases, and vice versa; such variation is due to the solubility of dissolved oxygen being inversely proportional to the temperature [29,30,31].

Regarding the repeatability test, the statistical analysis showed no significant differences among repetitions in any of the three parameters measured by the proposed LC OSH system. The mean temperatures were 23.66, 23.72, and 23.86 °C for repetitions 1, 2, and 3, respectively. For dissolved oxygen, the means were 6.44, 6.95, and 6.85 mg L^−1^ for the three repetitions; while, for pH, the mean values were 8.05, 8.37, and 7.94.

For the reliability test (precision, accuracy, and sensibility), the results showed a precision of 0.106 mg L^−1^ and a coefficient of variation (C.V.) of 0.42% for DO. For pH, the precision was 0.242 and the C.V. was 2.6%. In the case of temperature, the precision was 0.106 °C and the C.V. was 0.421%. Table 4 lists the results for the three parameters.

The above results indicate that the designed equipment can be used as a reliable tool in aquaculture, as the reads were very near to those considered as the true accepted value.

## 4. Discussion

The novelty of the proposed system mainly consists of the application of open-source hardware, which provides new opportunities for the research of in situ sensing in aquaculture systems with high precision and the behavior of cultured organisms (e.g., crustaceans, fish, and shellfish). The system allows for continuous recording and storing of water physicochemical variables for metrological traceability in precision aquaculture applications. The designed low-cost equipment can provide useful information, including under harsh aquaculture environments; therefore, more consideration in care and handling are required. A growing tendency in the development of instruments such as the proposed LC system to evaluate water quality has been observed worldwide, due to the problems faced by modern aquaculture; mainly those related to hygiene and environmental care. In developing countries, as is the case in Latin America, the innovation or development of different monitoring systems using open platforms (software or hardware) is of great interest, as they are cheaper [31] and can represent savings between 30–60% [26,30,32]; in addition, they have great potential to advance food security [33].

There are currently several ways to feed aquaculture crops, one of which considers the physicochemical variables of water—especially temperature and dissolved oxygen—as they have an effect on the food intake of organisms [34]. For this reason, different platforms have been developed to supply feed based on sensors [35,36,37] and so the proposed LC OSH monitoring system could be very useful for the development of automatic feeders. There are some very important challenges for the improvement of such systems, one of them being calibration: the programming codes could be made even more robust, such that the system can detect patterns associated with the need for re-calibration. The energy used for the evaluation of the proposed LC OSH system was provided by a local electricity supplier; however, some farms in the region do not have access to the electrical grid, due to their geographical location (i.e., difficulty of access or being a large distance from cities) [38] and, so, it would be appropriate to consider, in the future, powering these systems with solar energy or rechargeable batteries using solar energy. Another challenge is access to the Internet, as mentioned above. The locations of farms make Internet connection complicated although, due to technological advances in communications, there are now more and more tools available for connecting to the Internet [39], which would be excellent, for example, for adding the proposed LC OSH system to the IoT.

Some feasible improvements to the proposed LC OSH system could be: (1) the inclusion of a monitor of water-quality index, a tool which provides a general overview of the state of contamination of the water column, that can positively help to manage the farm for a better production response [4,5,39,40,41,42,43]; (2) the incorporation of more water-quality electrodes, such as conductivity, turbidity, and ion-selective electrodes; (3) furthermore, the addition of ion-selective electrodes (e.g., Mg, K, Ca, P, and Na) can be very useful in low-salinity cultures where the ionic balance is a key aspect—for example, some physiological and metabolic functions of shrimp, such as osmoregulation, have been directly related to Mg and K ions [44,45]; and (4) the inclusion of a module for the generation of early alerts, which are essential for farmers.

## 5. Conclusions

In this paper, the implementation of a low-cost open-source hardware system for monitoring and recording water-quality variables for aquaculture production was presented and evaluated. The proposed system was evaluated based on a comparative study assessing the measurement quality in terms of statistical analyses and comparison with commercial equipment. The results demonstrated that the data quality of the proposed system is appropriate for its use in aquaculture, with no statistically significant differences found between the proposed LC OSH system and a commercial system. The results obtained showed that the implemented system is not only a low-cost tool, but it is also precise and reliable. Considering the cost of the components, it also has a very good resolution and accuracy. These values can be used by researchers as a basis for developing remote-monitoring systems and automatic feeders based on sensors, as it is relatively easy-to-use measuring system for precision aquaculture with low cost, allowing its access to low-budget farmers.

## Figures and Tables

**Figure 1 sensors-20-06112-f001:**
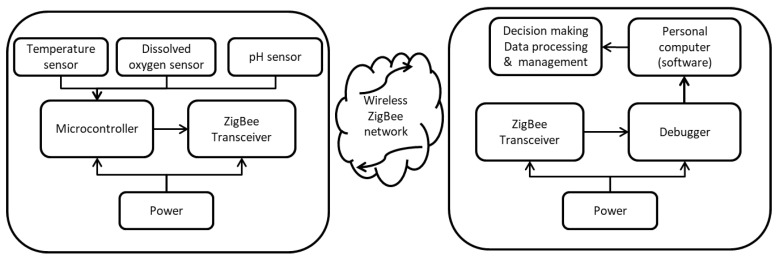
System component structure.

**Figure 2 sensors-20-06112-f002:**
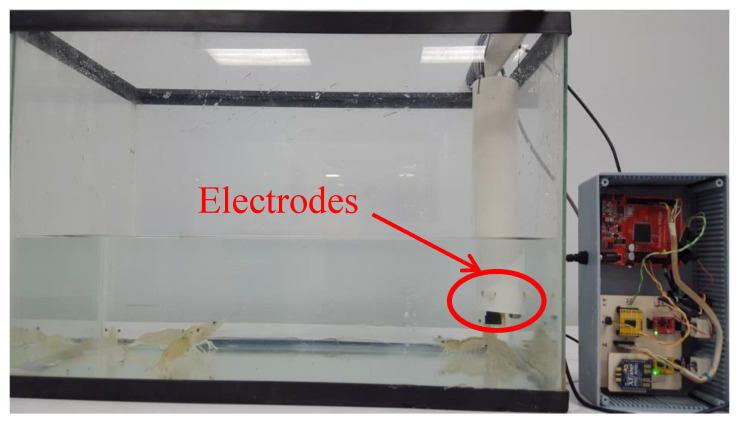
Experimental setup with 7 g five shrimp in 20 L of marine water and the proposed electronic system (gray box).

**Figure 3 sensors-20-06112-f003:**
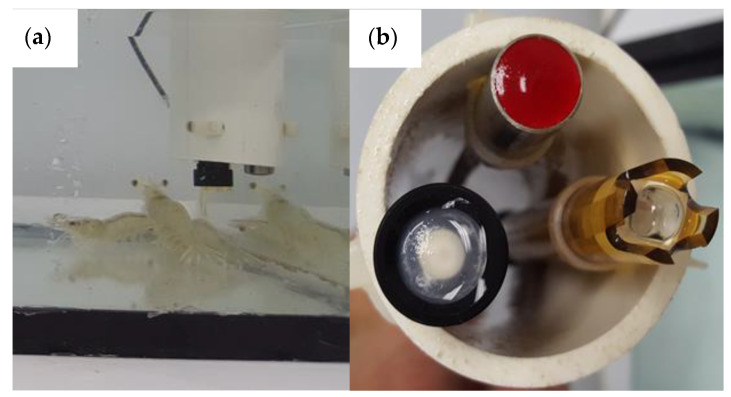
(**a**) PVC tube with vertical electrodes; and (**b**) underneath view of electrodes: black for dissolved oxygen, yellow for pH, and red sensor is temperature (from atlas scientific).

**Figure 4 sensors-20-06112-f004:**
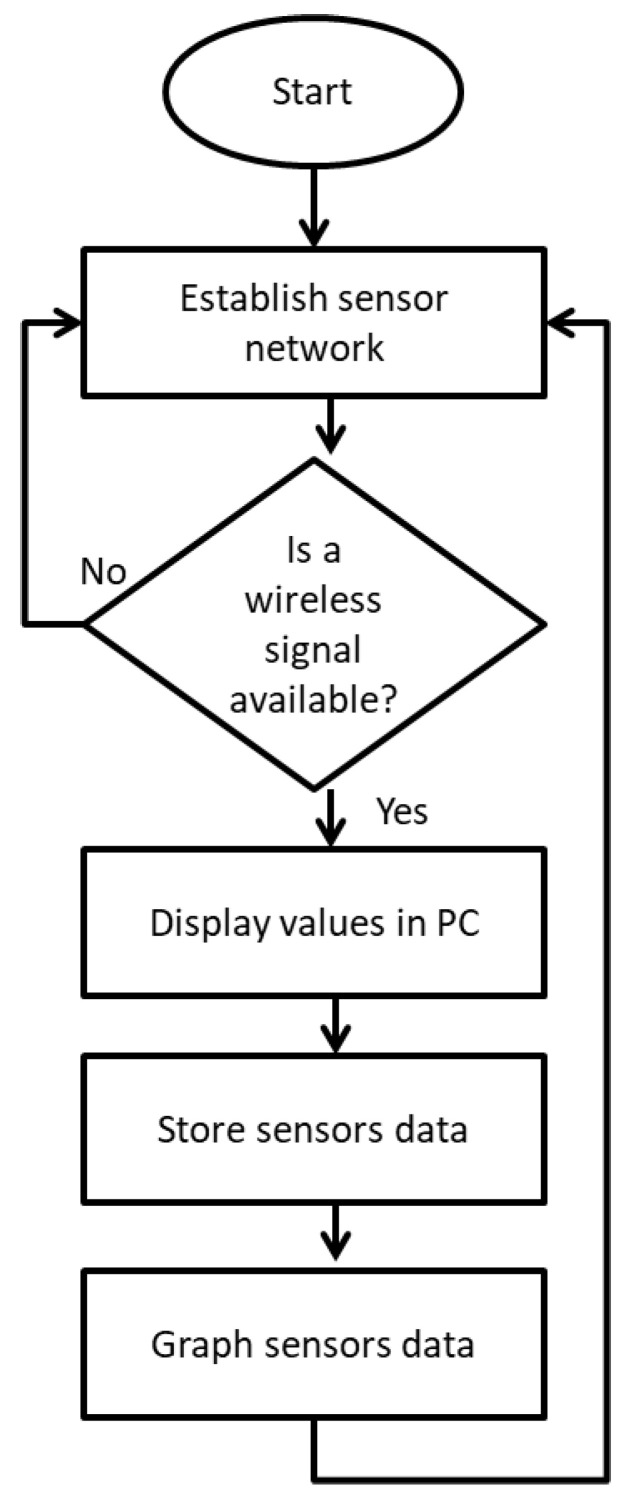
Flow diagram of communication software with low-cost open-source hardware (LC OSH).

**Figure 5 sensors-20-06112-f005:**
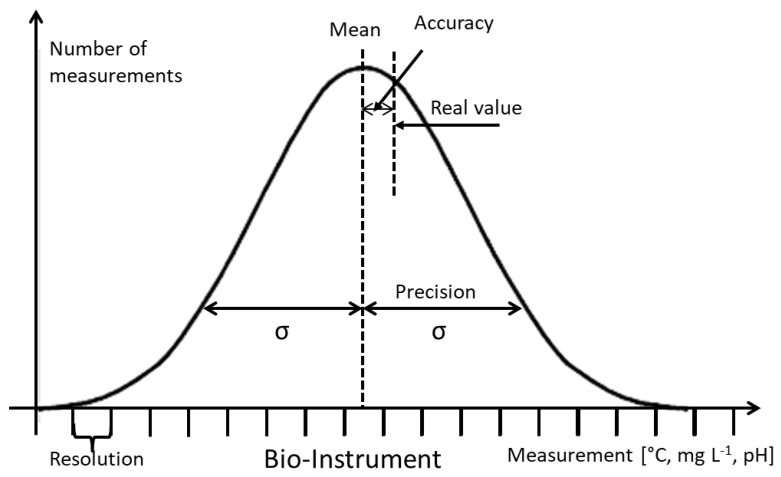
Normal distribution of instrument characteristics.

**Figure 6 sensors-20-06112-f006:**
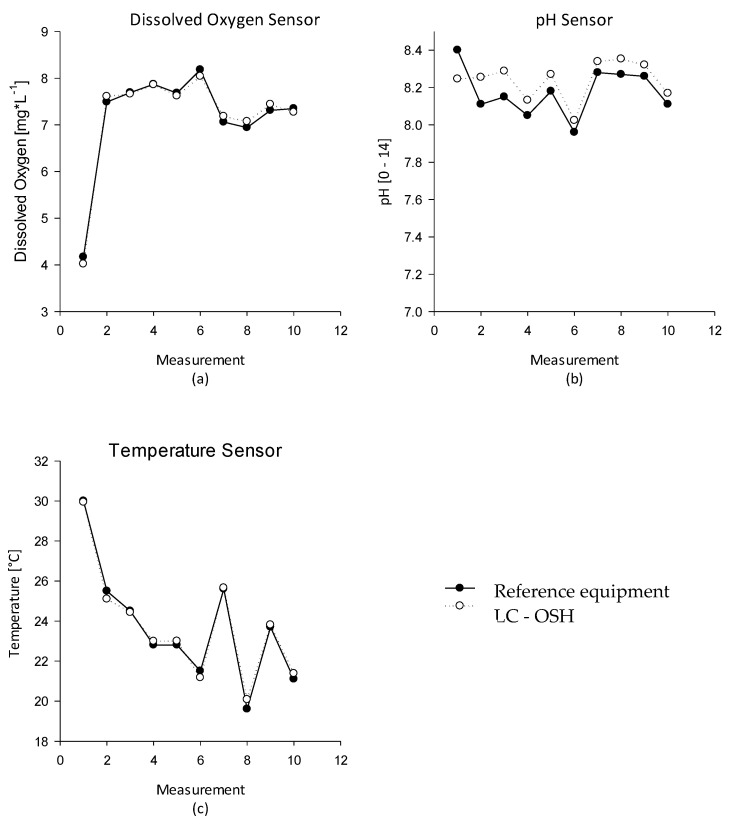
Comparison between reference equipment and low-cost proposed system: (**a**) dissolved oxygen sensor, (**b**) pH sensor, and (**c**) temperature sensor.

**Figure 7 sensors-20-06112-f007:**
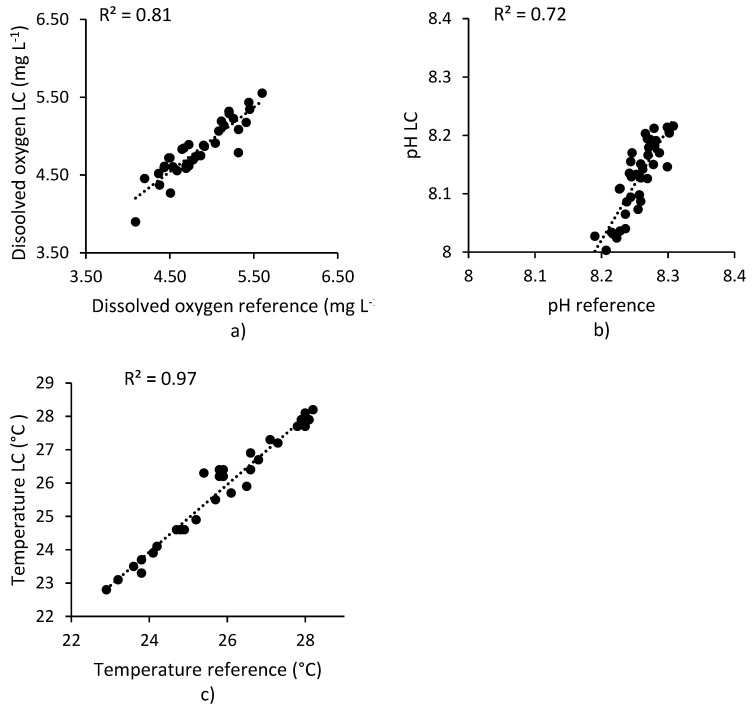
Scatterplots of low-cost open source hardware data versus reference equipment data. Dotted lines show the calculated r^2^ value of dissolved oxygen (**a**), pH (**b**) and temperature (**c**).

**Figure 8 sensors-20-06112-f008:**
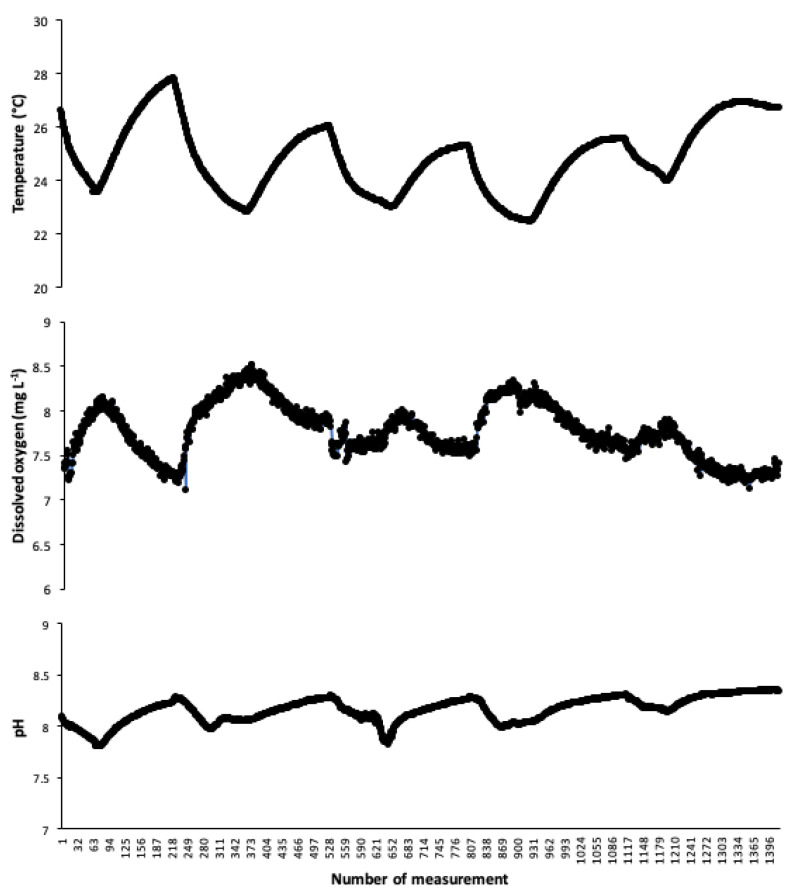
Behavior of the proposed low-cost system over the test time.

**Table 1 sensors-20-06112-t001:** Comparison of reads of water-quality parameters between the low-cost (LC) proposed system and reference equipment.

	Temperature [°C]		Dissolved Oxygen [mg L^−1^]		pH	
	Reference	LC	Reference	LC	Reference	LC
**Mean**	24.02	24.05	7.14	7.13	8.18	8.24
**Std. Dev.**	2.92	2.81	1.12	1.15	0.14	0.11
**C.V.**	12.31%	11.83%	15.59%	16.01%	1.71%	1.33%
**Min.**	19.60	20.08	4.17	4.02	7.96	8.02
**Max.**	30.00	29.94	8.81	8.04	8.40	8.35

**Table 2 sensors-20-06112-t002:** Comparison of temperature, dissolved oxygen, and pH in three repetitions. ^a^: statistically significant difference.

	Temperature [°C]			Dissolved Oxygen [mg L^−1^]			pH		
	LC 1	LC 2	LC 3	LC 1	LC 2	LC 3	LC 1	LC 2	LC 3
**Mean**	23.66 ^a^	23.72 ^a^	23.86 ^a^	6.44 ^a^	6.95 ^a^	6.85 ^a^	8.05 ^a^	8.37 ^a^	7.94 ^a^
**Std. Dev.**	0.248	0.271	0.251	0.104	0.080	0.066	0.119	0.056	0.055
**C.V.**	1.04%	1.14%	1.05%	1.61%	1.15%	0.96%	1.47%	0.66%	0.69%
**Min.**	23.35	23.26	23.56	6.20	6.77	6.73	7.873	8.345	7.891
**Max.**	24.11	24.15	24.36	6.62	7.11	6.98	8.199	8.415	8.059

**Table 3 sensors-20-06112-t003:** Price list of each of the components used.

Label	Type	Quantity	Approximate Price (€)
Power unit	9 V	1	4
Transceiver modules	XBEE XBP24-BZ7UIT-004	2	72
Microcontroller	Iteaduino MEGA2650	1	13
Debugger	USB FOCA FT232RL	1	7
Serial port expander	74HC4052 Multiplexor	1	10
Embedded dissolved oxygen circuit	Atlas Scientific	1	39
Dissolved oxygen probe	Membrane-type PTFE	1	200
Embedded pH circuit	Atlas Scientific	1	35
pH probe	Silver/silver chloride	1	70
Total			450

**Table 4 sensors-20-06112-t004:** Reliability (precision, accuracy, and sensibility) of the proposed low-cost water-quality system (σ is standard deviation and C.V. coefficient of variation).

	Temperature	Dissolved Oxygen	pH
**Precision [σ, C.V.]**	±0.106 °C, 0.42%	±0.177 mg L^−1^, 4.28%	±0.242, 2.6%
**Accuracy**	0.18 °C	0.016	0.006
**Sensibility**	0.038 °C	0.017 mg L^−1^	0.018

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
