# Peer review of "Implementation and Evaluation of Open-Source Hardware to Monitor Water Quality in Precision Aquaculture"

_sensors, 2020, doi:10.3390/s20216112_

Round 1

Reviewer 1 Report

Review of the Manuscript ID sensors-830265

Revised on June, 1st 2020

Title:    Implementation and evaluation of an open source hardware to monitor water quality in precision aquaculture

Authors:          Rafael Apolinar Bórquez López, Luis Rafael Martinez Cordova, Juan Carlos Gil Nuñez, Jose Reyes Gonzalez Galaviz, Jose Cuauhtemoc Ibarra Gamez, Ramon Casillas Hernandez *

The authors present a design of a low-cost monitoring system based on an open source platform for monitoring of water quality parameters in aquaculture. The topic is novel and remote sensing and implementation of IoT methodologies is of the utmost importance in Precision Aquaculture advancement. The manuscript offers a standardization procedure for the low-cost monitoring system and a low-cost open source hardware through comparison to the reference equipment. The presented results can be a valuable information for researchers dedicated to development of the remote sensing systems and optimizing the costs of aquaculture production. However, I do have some comments:

Comment 1: The English language is very problematic throughout the whole text in terms of spelling, grammar as well as syntax errors. The text should be edited by an English-speaking person. Some of my spelling corrections are posted as notes in the original pdf file, however the whole manuscript needs a language revision and rewriting.

Comment 2: There is some confusion and repeating in the Introduction part within lines 39-50. This section should be rewritten.

Comment 3: The choice of the low cost equipment, its assembly and logic behind should be better explained and justified in the framework of the scientific aim (lines 77-80), as part of the methodology (operation algorithm) and in the discussion part of the manuscript.

Comment 4: Fig. 3 is missing. Please provide Fig. 3 or correct the numbering of the figures.

Comment 5: Fig. 6 there are typographic errors in the explanations given on graphical presentation (e.g. ? )

Comment 6: A whole chapter describing part of the results is written in Spanish. Lines 283-291.

Comment 7: Lines 335-343 have appropriately tackled the relevant issues of water quality in precision aquaculture and in particular the parameters for which the currently available equipment does not provide sufficiently precise monitoring.  However, the challenges, as well as the added value of the related research efforts needs to be discussed a bit more into depth.

Comment 8: The conclusion Lines 352-357 should be rewritten pointing out the added value of the obtained results in sense of further development of the precise aquaculture remote monitoring systems and automation control.

Comment 9: There are many incorrectly cited references:

ref. No. 3 and 30: the source is missing

ref. No. 6: the title of the article is missing

ref. No. 10: the title is incomplete and the journal name is missing

ref. No. 26: same reference as 32

ref. No. 29: it is not clear if this is a book chapter or a research article, there are no editors and publisher, or journal name

ref. No. 34, 35, 36: full name of the journal is missing

Reviewer 2 Report

The article is certainly interesting for Sensors readers. Provides a relatively easy measuring system for aquaculture farms.

Abstract introduces well to the problems of work.

Inroduction

- I propose to add that the range of tested water parameters is related to the individual characteristics of both water and cultured organisms.
- I suggest moving to this section the literature reviews contained at the beginning of sections 2.2.1, 2.2.2 and 2.2.3,
- the purpose of the work is clear.

Materials and Methods

- in the abstract, the authors write about real time system (line 20), while on line 91 with a 10-minute measuring interval of the system,

- Line 98 - I suggest a simple literature reference instead of the company's website. Similarly, line 141 and signature under Fig. 4. I suggest moving the website address to the bibliography. This will avoid any suspicion of advertising.
- Signature Fig. 4 - doubled "electrodes",

- the following information is missing: how long did the measurements take, and how often they were carried out?
- was water mixing used at the measuring station?
Section 2.2.1 begins with a literature review. In my opinion, the text should start from line 133

Line 136-137 - measuring range from 0.01 with a precision of 0.20 mg / L means low accuracy at low concentrations
Sections 2.2.2 and 2.2.3 - as above.
Fig. 6 - some "rubbish" in the descriptions
Equation 1-6 - no reference in the text. Are they needed in the paper?

Results

Tab. 2 - Can one infer statistically using three measurements (no previous information on their full number)? What does the letter "a" mean by numbers (Mean line)?
Line 249 - it would be worth adding the cost of commercial software

Tab. 3 should in my opinion be in section 2 when describing the methods
Line 277 record units, line 281- record r2
Line 283 and beyond - repetition in Spanish,

Fig 9 – zastosowanie skali czasu na osi poziomej byłby bardziej czytelne moim zdaniem,

The 312 reliability contains also a lifetime of devices - this has not been studied
Discussion
Listing potential expansion directions - it is worth referring to the first comment from the Introduction section

References – pos. 21 – format

Round 2

Reviewer 1 Report

Review of the Manuscript ID sensors-830265

Revised on July, 19th 2020 

Title:    Implementation and evaluation of an open source hardware to monitor water quality in precision aquaculture

Authors:           Rafael Apolinar Bórquez López, Luis Rafael Martinez Cordova, Juan Carlos Gil Nuñez, Jose Reyes Gonzalez Galaviz, Jose Cuauhtemoc Ibarra Gamez, Ramon Casillas Hernandez *

General comment about the revised manuscript ID sensors-830265

The authors have improved the manuscript in regard to most of my remarks. However the English style and writing still need to be substantially improved in order to allow for the acceptance of the manuscript in Sensors. I have tried to help them in that respect, however I still think the manuscript should be checked by the English native speaking scientist.

Please find bellow the speciffic remarks:

Line 50: ...to maintain adequate conditions and avoid undesirable situations which may lead to the collapse of aquaculture systems...

Line 52: ... abrupt variations of temperature...

Line 64: ... may result. If the level decreases under 4, the acid death is produced.

Line 67 ... varies...

Line 69 ... metabolism...

Line 95: ...that is why we decided to perform this study...

Line 104: ... system was designed and evaluated...

Line 108: ...In Briefly, the system consists of a hardware development platform at open code (word order!)…

Line 110 … the system is adapted to the software on a desktop computer through a serial port based on a depurator

Line 119 …The plates

Line 136 …comparing them with to the commercial equipment

Line 138 …installed completely vertical…

Line 159 …It needs to be calibrated once a year, depending on the operation conditions and time of use

Line 163-164 …Analogue passive pH electrode was connected to embedded circuit used for monitoring the pH in the system, similar to that used for DO;

Line 173 … to ensure more precise reads

Line 186 …responsible for calling them and establishing a database of water quality

Line 206 …comparison of the random results

Line 207 …by means of ANOVA

Line 211 …the basis

Line 212 …the reproducibility

Line 216… sensitivity analysis

Line 250 …reproducibility test

Line 271 …database

Line 275 … gives

Lines 275-279: “The comparison between reference equipment against LC - OSH follow same pattern, the difference in the median values between groups are not great enough to exclude the possibility that the difference is due to random sampling variability, there is no statistically significant difference (P < 0.05) determined by Student ́s test.”

I guess the authors tried to say:

The data obtained by LC-OSH system compared to reference equipment follow the same trend and demonstrate no statistically significant difference (P < 0.05) of the median values between groups, determined by Student’s test. The possibility should not be excluded that the difference is caused by random sampling variability.

Lines 284-291 The sentence should be broken into several shorter (2-4) and easier to follow sentences.

Line 289 … in a similarly way

Line 303 …the maximum values

Line 304 … “It can be noted…” should be formulate4d as separate self-standing sentence

Lines 306-308 It can be noted that as temperature increases, oxygen concentration decreases and vice versa, as a consequence of inverse dependence of oxygen solubility in respect to temperature [29-31].”

Line 313 Regarding to the repetitiveness test

Lines 332-333 …for in situ sensing research in high precision aquaculture systems and behavior of culture organisms

Lines 366-367 …the incorporation of more electrodes for water quality assessment, such as conductivity, turbidity and ion selective electrodes

Line 370 functions

Line 381 statistical ly

Line 370 …The results obtained shows

Line 384 … These values

Line 386 … Since it is relatively simple and user-friendly measuring system for aquaculture, it can also be at the reach of the low budget farmers.

Best regards.

Author Response

Reviewer suggested that the manuscript should be checked by the English native speaking scientist. We follow reviewer suggestion, this will help us to have a clearer manuscript.
